# Differential Diagnosis between Oral Metastasis of Renal Cell Carcinoma and Salivary Gland Cancer

**DOI:** 10.3390/diagnostics11030506

**Published:** 2021-03-12

**Authors:** Yoshihiro Morita, Kana Kashima, Mao Suzuki, Hiroko Kinosada, Akari Teramoto, Yuka Matsumiya, Narikazu Uzawa

**Affiliations:** Department of Oral and Maxillofacial Surgery II, Graduate School of Dentistry, Osaka University, 1-8 Yamadaoka, Suita-shi, Osaka 565-0871, Japan; k-kashima@dent.osaka-u.ac.jp (K.K.); mmaommao0429@dent.osaka-u.ac.jp (M.S.); h-kinosada@dent.osaka-u.ac.jp (H.K.); a-teramoto@dent.osaka-u.ac.jp (A.T.); yuka_loves_tigers@dent.osaka-u.ac.jp (Y.M.); uzawa@dent.osaka-u.ac.jp (N.U.)

**Keywords:** renal cell carcinoma, salivary gland cancer, clear cell, differential diagnosis

## Abstract

Renal cell carcinoma, which has clear cells in 70% of cases, has a high frequency of hematogenous distant metastases to lung, bone, liver, and other areas. Metastatic cancer accounts for 1 to 3% of malignant tumors in the stomatognathic region, and the metastasis of renal cell carcinoma to the oral mucosal tissue, though extremely rare, does occur. In addition, clear cells have been observed in some salivary gland cancers in the oral cavity. Therefore, the differential diagnosis of metastatic renal cell carcinoma and salivary gland cancer is important. This review discusses the differential diagnosis between metastatic renal cell carcinoma and malignant tumors of the salivary gland.

## 1. Introduction

The oral cavity is an extremely rare site of cancer metastasis. Metastases account for approximately only 1% of all malignant tumors in the oral cavity [1]. When primary tumors metastasize to the oral cavity, the most common primary sites are lung carcinoma in males and breast carcinoma in females, followed by renal cell carcinoma (RCC) [2]. In RCC cases, approximately one third of the cases develop metastases. Distant metastases are detected in approximately half of RCC metastases cases after the initial diagnosis. The most common sites of distant metastases from RCC are the lungs, bone, liver, adrenal glands, contralateral kidney, and brain [3]. Although clear cells, which are either epithelial or mesenchymal cells consisting of pale or clear cytoplasm that have a distinct nucleus, can be observed in both benign and malignant epithelial, mesenchymal, melanocytic, and hematopoietic tumors, they are relatively rare in the head and neck region [4]. Because they share similar histological features, it is difficult to make definitive diagnosis in many of these tumors when clear cells predominate [5]. Focal clear cell changes may appear secondarily, reflecting clonal evolution, or may become more extensive with tumor progression [4]. The diagnosis of clear cell tumors may be made challenging by these changes.

## 2. Renal Cell Carcinoma

Renal cell carcinoma (RCC) accounts for more than 90% of cases of malignant renal disease and is thus the most common form. The incidence of RCC is high worldwide; it is the sixth most common cancer in men and the 10th most common in women [6], with a male–female ratio of 1.5:1. The peak incidence of RCC occurs in patients aged between 60 and 70 years.

There are several recognized histologic variants of RCC. Clear cell RCC accounts for 75% of cases, papillary RCC accounts for 10 to 15%, and chromophobe RCC accounts for 5%. These three histologic variants represent 90% of all RCCs [7]. Several risk factors that favor RCC development have been reported; these include a high body mass index [8], urinary stones in male patients [9], type 2 diabetes mellitus in female patients [10], chronic liver and kidney diseases [11], and the long-term use of analgesics [12], as well as a number of environmental factors [13]. As RCCs are frequently found in the course of diagnostic investigations for other purposes, they have been increasingly diagnosed in recent years.

The prognosis for RCC varies because it depends on the anatomical, histological, clinical, and molecular features of the disease. Additionally, in the genomic features, 9p loss appears a reliable and promising marker of patients with renal cell carcinoma associated with a worse prognosis [14]. Overall, the 5-year survival rate for RCC is 74%, but is lower in patients with stage III locoregional disease (54%). Moreover, the survival rate decreases to 8% in patients with metastatic disease [15].

## 3. Classification of Clear Cell Tumors in the Oral Cavity

Physiologically and pathologically, clear cells can be classified broadly. Pathological clear cells are encountered in a variety of tumors, and physiological clear cells, which can either be epithelial or mesenchymal in origin, include both odontogenic and nonodontogenic tissues. It can therefore be said that clear cell tumors constitute a heterogeneous group of lesions.

### 3.1. Clear Cell Metastatic Tumors

Carcinomas from the kidney, liver, large bowel, prostate, and thyroid can metastasize to the maxillofacial area and are known to have the potential for clear cell differentiation; RCC does so most frequently [5]. RCC metastatic lesions can sometimes mimic salivary gland and odontogenic tumors, but are characterized by hemorrhagic areas and the prominent sinusoidal vascular component [16]. The prognosis of patients with metastatic carcinomas tends to be poor.

Local or distant metastases are known to occur in about one third of patients, and recurrence is seen in approximately 25% of patients with localized renal disease who undergo nephrectomy. RCC metastatic lesions typically develop in the lungs, regional lymph nodes, liver, bones, and brain [17].

Although relatively rare, RCC metastatic lesions are sometimes found in the oral cavity, predominantly involving the tongue, followed by the gingiva and maxillary bones [18,19].

### 3.2. Clear Cell Salivary Gland Tumors

Although clear cell salivary gland tumors, which constitute less than 1% of all primary salivary gland tumors [20], are often malignant in nature, they include two types of benign lesions: clear cell variants of oncocytoma and myoepithelioma. Tumors that consist exclusively of myoepithelial cells are designated as myoepitheliomas, with both benign and malignant variants containing clear cells, although the latter is less common. Most malignant myoepitheliomas are less monomorphic than benign myoepitheliomas [21]. Epithelial myoepithelial carcinoma derives from intercalated ducts with biphasic duct-like structures. In acinic cell carcinoma, clear cells tend to occupy only a small portion of the tumor, whereas clear cell mucoepidermoid carcinoma can be easily identified by an admixture of clear squamous, mucous, and intermediate cells [16]. In addition, hyalinizing clear cell carcinoma demonstrates tumor cell clusters separated by broad bands of hyalinized stroma that have been shown to undergo myxoid or hyaline degeneration [16].

Clear cell metastatic tumors should be distinguished from primary salivary tumors because of their diagnostic and therapeutic importance [4]. In terms of biological behavior, clear cell salivary gland tumors are considered low-grade malignancies. This is because, in spite of their benign appearance, clear cell salivary gland tumors are capable of locally infiltrative growth and destruction, as well as metastasis, and tend to be associated with a poor prognosis [20].

### 3.3. Other Clear Cell Lesions in the Oral Cavity

Odontogenic neoplasms with significant clear cell components are exceedingly rare [22]. Most of these odontogenic lesions are composed of areas with characteristic histological features, which helps to distinguish them from each other [23].

Surface epidermal and cutaneous adnexal tumors are frequently found in the facial skin. Squamous and basal cell carcinomas have both been observed to show clear cell variants, with 90% of such neoplasms being found in the skin of the head and neck. Although individual islands within these neoplasms tend to consist of solely clear cells, adjacent cells have been reported to exhibit the typical characteristics of either basal or squamous cell carcinoma [23].

The incidence of lipoma in the oral cavity is known to be about 1% [24]. In addition, liposarcoma accounts for about 5.6 to 9% of cases in the head and neck area [23]. Histologically, lipomas are characterized by mature adipose tissue without cytologic atypia [24]. On the other hand, liposarcoma shows the presence of lipoblasts, cellular pleomorphism, vascular proliferation, and mitotic activity. The prognosis of liposarcoma of the oral cavity is generally good owing to the small size of these neoplasms and the predominance of both myxoid and well-differentiated types [23].

Malignant melanomas and melanocytic nevi are composed of cells with different morphological phenotypes. Balloon cell melanoma, a rare form of vertical proliferative melanoma, is characterized by the nodular growth of neoplastic balloon cells. These tumors, which are characterized by large cell nests and sheets with clear or finely vacuolated cytoplasm, have a similar prognosis to other types of melanoma. In the largest series recorded, 57.5% of patients died of metastatic disease from 2 months to 12 years after initial surgery [23]. The clear cell variant of chondrosarcoma is very rare, accounting for only about 2% of all chondrosarcomas, with only nine cases having been reported in the head and neck area. In cases involving the head and neck, the histological findings have been consistent with the general characteristics of this entity throughout the body, and represent a component of conventional chondrosarcoma. The clear cell variant of chondrosarcoma is a distinct low-grade sarcoma with potential for local recurrence or distant metastasis, with a recurrence rate of approximately 20% reported in several case series [25].

## 4. Differential Diagnosis of Clear Cell Tumors in the Oral Cavity

Differentiating among clear cell tumors histologically is difficult by conventional light microscopy alone. This is especially true when trying to distinguish RCC metastases from clear cell malignancies of the salivary glands. The differential diagnosis between malignant salivary gland tumors including clear cells and metastasis of RCC is made using special staining or immunohistochemical staining based on histological characteristics. The major components of salivary gland tumors, which are characterized by clear cell changes, are contributed by the myoepithelial cells [26]. Based on this, primary salivary clear cell neoplasias can be divided into those that diagnostically require evidence of myoepithelial differentiation (myoepithelioma or myoepithelial carcinoma and epithelial myoepithelial carcinoma) and those that do not. Clear cell variants of salivary gland tumors, such as clear cell carcinoma, acinic cell carcinoma, mucoepidermoid carcinoma, and oncocytoma, do not have myoepithelial differentiation. In this regard, calponin-based immunohistochemistry can be an important diagnostic tool [26]. A chart for the differential diagnosis of clear cell malignant tumors is proposed in Figure 1. Schmidt et al. mentioned that the clinical usefulness of fine-needle aspiration cytology for the diagnosis of salivary gland lesions is controversial in their systematic review paper [27].

### 4.1. Characteristic Histopathological Findings for Making the Diagnosis

Clear cell carcinomas of the salivary glands are usually seen as nests of clear cells divided by thin, fibrous connective septa and irregular vascular tissue. Most malignant myoepitheliomas frequently have high mitotic activity and atypical forms [28]. Table 1 shows the histological features useful for the differential diagnosis of salivary gland tumors containing malignant clear cells and metastatic RCC.

### 4.2. Special Stains for Making the Diagnosis

#### 4.2.1. Periodic Acid-Schiff Stain

Periodic acid-Schiff (PAS) stain is positive with glycogen, some mucins, and mucopolysaccharides, which can be diastase-sensitive or resistant. PAS diastase stain is a PAS stain used in combination with diastase, an enzyme that breaks down glycogen [26].

#### 4.2.2. Mucin Stains

Neutral and acid-simple non-sulfated and acid-complex sulfated mucins and mucopolysaccharides are stained by a PAS stain, but not acid-simple mesenchymal mucins and acid-complex connective tissue mucins. Alcian blue is used to detect neutral and acidic mucopolysaccharides, sialomucin, and sulfomucin. Mucicarmine stains acidic mucin [26].

#### 4.2.3. Oil Red O and Sudan Black B Stains

Oil red O and Sudan black B stain neutral lipids in frozen sections and lipoproteins in paraffin sections, and a combination of PAS with Alcian blue is a pan-mucin marker. Special stains used for the detection of glycogen and lipids in the cytoplasm of RCC cells are PAS, Oil red O, and Sudan black B [26].

### 4.3. Immunohistochemical Stains for Diagnosis

Immunohistochemical staining can assist in the diagnosis because RCC metastases show a strong reaction to vimentin, as well as focal cytokeratin (CK) positivity; by contrast, minor salivary gland cancers show diffuse CK positivity [12]. In addition, immunohistochemical staining for p63 has been shown to be useful for distinguishing mucoepidermoid carcinoma from some clear cell tumors [29], and Cluster of Differentiation (CD) 10 expression in RCCs are useful as a marker in the differential diagnosis of several tumors. Moreover, CD13 and GATA binding protein 3 (GATA-3) immunostains may serve as a diagnostic aid in differentiating subtypes of RCC [30].

#### 4.3.1. Myoepithelial Markers

In addition to actin and S-100 [31], which are considered the classical myoepithelial markers, a number of myoepithelial markers are considered useful for immunohistochemical studies of formalin-fixed, paraffin-embedded tissues because they can recognize a variety of antigens in myoepithelial cells and have shown good sensitivity and specificity [31]. Among these myoepithelial markers, calponin, p63, CD10, and high-molecular weight cytokeratins (HMWCK; CK5/6 and CK14) have been confirmed to be effective in routine clinical practice because of their sensitivity, specificity, and ease of interpretation [31]. The utility of *α* smooth muscle actin (*α*-SMA), S-100, calponin, p63, CD10, CK5/6, and CK14 for identifying the neoplastic myoepithelial components of adenomyoepithelioma has been reported by Moritani et al. [31], who concluded that all seven myoepithelial markers were useful for the diagnosis of adenomyoepithelioma. It is reasonable to select one or more markers from each group for the immunohistochemical panel of adenomyoepithelioma in consideration of the relatively high sensitivity of *α*-SMA, calponin, and p63 in both spindle and clear cell lesions, as well as the unique paradoxical staining pattern and relatively high sensitivity of HMWCK in spindle cell lesions. When selecting immunohistochemical markers, taking the morphology of the lesion (i.e., spindle or clear cell) into account is also important.

Glial fibrillary acidic proteins (GFAPs) have also been reported to be sensitive markers of myoepithelial differentiation in salivary lesions [32].

#### 4.3.2. Epithelial Markers

Both pancytokeratin (panCK) (AE1/AE3) and Epithelial membrane antigen (EMA) are epithelium-specific antibodies, and as the basic components of the cellular structure of normal epithelial and epithelial cancer cells, are frequently used to differentiate tumors in terms of whether they originate from the epithelium [33].

#### 4.3.3. Other Markers

Although vimentin has commonly been used as a specific marker for neoplastic myoepithelial cells [34], its specificity has recently been reconsidered [35]. According to the results in this study, vimentin and calponin appear to be sensitive immunohistochemical markers of myoepithelial cells in salivary gland tumors, as they exhibit a predominantly intense expression. Therefore, the combination of calponin and vimentin is recommended for the identification of myoepithelial cells in neoplasms, as these cells can be found in various stages of differentiation [36].

Renal cell carcinoma marker (RCC-Ma) is a monoclonal antibody against a normal renal proximal tubule antigen. RCC-Ma expression is relatively specific for primary clear cell RCC [37]. Additionally, paired box 8 (PAX-8) and Carbonic anhydrase 9 (CA9, CAIX) expression are useful diagnostic markers for RCC [38,39]. PAX-8 expression is detected in the primary tumor and distant sites. Clear cell RCC has lower PAX-8 expression and is less frequently positive compared with normal tissue and other histological types. Therefore, the lack of expression does not exclude a tumor of renal origin. CA9 is not expressed in normal renal tissue but is expressed in most clear cell RCC through hypoxia inducible factor (HIF) -1α accumulation driven.

## 5. Review of the Literature for the Diagnosis of RCC Metastasis

### 5.1. Search Strategy

A literature search was performed to retrieve previous studies describing the occurrence of oral metastases of renal cancer in the last 10 years. The following search items, combined with the Boolean term “AND,” were used to perform an electronic search in the PubMed database: oral metastasis, renal cell carcinoma, case report. Only English ones were selected from the literatures obtained by this search.

### 5.2. Review of the Literature

In total, 65 articles published between 2001 and 2020 were found (Table 2); 49 (74.2%) of the 66 cases reported were male (male-female ratio, 2.9:1), and patient ages ranged between 31 and 89 years (mean age, 52.6 years).

A total of 35 of the 66 cases were previously diagnosed with RCC and had a history of treatment. The term from the initial diagnosis of RCC was up to 26 years, averaging 6.1 years. On the other hand, 26 cases were diagnosed with RCC in the kidney after the lesions in the oral cavity were found.

The tongue was the most common metastatic site in the oral region (33.3%), followed by the parotid glands (22.7%) and gingiva (21.2%). Although only a few cases have been reported to have metastasized to bone (4.5%), it is possible that some of the cases that metastasized to the gingiva included metastasized cases to the bone.

Immunohistochemical staining was performed in 39 cases for the diagnosis of metastatic lesions of RCC. It was suggested that immunohistochemical staining is not always necessary to diagnose RCC metastases, since the remaining 27 cases were diagnosed without immunohistochemistry. Among the cases that underwent immunohistochemical staining, vimentin, and CD10 were tested in many cases (61.4 and 64.1%, respectively). In addition, the results of staining were positive in all of those cases. S-100 protein was tested in 35.9% of cases: one case was positive, and all others were negative. It appears that these immunohistochemical stains are useful for diagnosing RCC metastases.

## 6. Conclusions

It is known that RCC rarely metastasizes to the head and neck. Therefore, in patients with a history of RCC, metastatic RCC should be considered in the differential diagnosis of oral lesions. In addition, it should be considered that RCC may be found by whole body evaluation, such as positron emission tomography-computed tomography, even in patients with no history of RCC. In patients with a clear cell malignant tumor in the oral cavity, immunohistochemical staining is important to differentiate between metastatic RCC and malignant tumors of salivary gland origin.

## Figures and Tables

**Figure 1 diagnostics-11-00506-f001:**
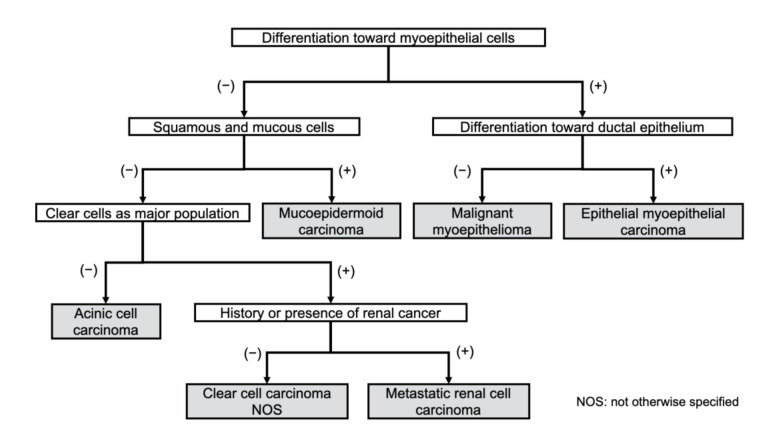
Diagnostic work chart of clear cell malignant tumors of the oral region.

**Table 1 diagnostics-11-00506-t001:** Diagnostic immunohistochemical markers, special stains, and histopathological features for clear cell tumors.

Name	Immunohistochemistry	Special Stains	Histopathology
Malignant myoepitheliomas	Myoepithelial markers (+), p63(+)	PAS (+), Mucin stains (−)	-
Epithelial myoepithelial carcinoma	Epithelial markers (+: inner cells), Myoepithelial markers (+: outer clear cells)	PAS (+), Mucin stains (−)	Biphasic tubular structure-outer clear and inner cuboidal cells
Mucoepidermoid carcinoma	Myoepithelial markers (−), p63(+)	PAS (+), Mucin stains (+)	Mucus and intermediate cells
Acinic cell carcinoma	Amylase (+)	PAS (+) *	Granular cytoplasm
Clear cell carcinoma	Epithelial markers (+), Myoepithelial markers (−)	PAS (+), Mucin stains (−)	Hyalinizing stroma
Renal cell carcinoma	CD10(+), vimentin (+), RCC marker (+)	Oil red O stains (+), Sudan black B stains (+)	Highly vascular stroma, sinusoidal spaces

Mucin stains: alcian blue, mucicarmine; Myoepithelial markers: SMA, S-100, CK, GFAP, vimentin, calponin; Epithelial markers: panCK and EMA; * PAS diastase stain.

**Table 2 diagnostics-11-00506-t002:** Cases of metastases of renal cell carcinoma to the oral region: Review of the literature.

Metastasis Site	Age (y)	Sex	Time from Diagnosis of RCC	Special Stains	IHC Markers	Year	Ref.
Parotid gland	64	M	10 years	Not performed	vimentin (+), keratin (+), CEA (+)	2001	[40]
Parotid gland	61	M	Same time	Not performed	Not performed	2001	[41]
Parotid gland	83	F	10 years	Not performed	vimentin (+), keratin (+), CEA (−)	2002	[42]
Lip	70	M	Unknown	Not performed	vimentin (+)	2002	[2]
Tongue	58	M	Unknown	Not performed	Not performed	2002	[2]
Tongue	45	M	8 weeks	Not performed	Not performed	2003	[43]
Tongue	62	M	Same time	Not performed	AE1/AE3 (+), S100 (−), CEA (−)	2003	[44]
Tongue	87	F	10 years	PAS (+)	CK (+), vimentin (+), S100 (−), MSA (−), desmin (−), CEA (−), HMB45 (−), CD31 (−)	2004	[12]
Parotid gland	59	F	10 years	PAS (+)	CK (−), vimentin (+), EMA (+), CEA (−), SMA (−), S100 (−)	2004	[45]
Parotid gland	67	M	Same time	Not performed	LMWCK (+), vimentin (+), CD10(+), S100(−), CK7(−), CK20(−), EMA (−)	2005	[5]
Tongue	49	F	6 months	PAS (+)	Not performed	2006	[46]
Parotid gland	74	F	7 years	Not performed	panCK (+), vimentin (+), S100 (−), SMA (−), CK7 (−), CK20 (−)	2007	[47]
Parotid gland	58	F	Same time	Not performed	CD10 (+), PNRA (+), vimentin (+), AE1/AE3 (+), calponin (−), SMA (−), S100 (−), CK7 (−)	2008	[48]
Parotid gland	76	F	9 years	Not performed	CD10 (+), PNRA (+), vimentin (+), CK7 (−)	2008	[48]
Parotid gland	62	M	5 years	Not performed	CAM5.2 (+), vimentin (+), CK18 (+), CK8 (+), CK7 (−), PNRA (−)	2008	[48]
Tongue	78	M	Same time	Not performed	Not performed	2008	[49]
Tongue	63	M	4 years	Not performed	vimentin (+), CD10 (+), GCDFP (−), S100 (−), HMB45 (−), MSA (−), desmin (−)	2008	[3]
Gingiva	74	M	Same time	Not performed	Not performed	2008	[50]
Parotid gland	69	M	1 year	Not performed	Not performed	2008	[51]
Gingiva	63	M	2 years	Not performed	Not performed	2009	[19]
Gingiva	52	M	Same time	Not performed	Not performed	2009	[1]
Buccal mucosa	74	M	1 year	Not performed	Not performed	2010	[52]
Tongue	47	M	Same time	Not performed	CD10 (+), EMA (+), AE1/AE3 (+)	2011	[53]
Tongue	48	F	3 years	Not performed	Not performed	2011	[54]
Gingiva	63	M	2 years	Not performed	CD10 (+)	2012	[55]
Tongue	72	M	Same time	Not performed	Not performed	2012	[56]
Submandibular gland	60	M	9 years	Not performed	Not performed	2012	[57]
Tongue	48	M	5 years	Not performed	Not performed	2011	[58]
Parotid gland	82	M	20 years	Not performed	Not performed	2012	[59]
Lymph	56	F	6 years	Not performed	Not performed	2011	[60]
Tongue	64	F	14 years	Not performed	AE1/AE3 (+), EMA (+), vimentin (+), CD10 (+), CK7 (−), CK19 (−), CK20 (−), TTF1 (−)	2012	[61]
Parotid gland	79	F	16 years	Not performed	Not performed	2012	[62]
Tongue	70	M	16 years	Not performed	CD10 (+)	2012	[63]
Tongue	66	M	Same time	Not performed	Not performed	2013	[64]
Parotid gland	44	F	7 months	Not performed	Not performed	2013	[65]
Tongue	65	M	2 months	Not performed	Not performed	2013	[66]
Tongue	65	M	Same time	Not performed	Not performed	2013	[66]
Mandible bone	57	M	Same time	Not performed	desmin (+), CK (+), vimentin (+), S100 (−), HMB45 (−)	2013	[67]
Parotid gland	64	M	6 years	PAS (+)	AE1/AE3/CAM5.2 (+), vimentin (+), RCC (+), CD10 (+), CA9 (+), CK7 (−), CK20 (−), calponin (−), p63 (−)	2014	[68]
Tongue	80	M	4 years	Mucicarmine (+), PAS (+)	CD10 (+), CK7 (−), Melan-A (−)	2014	[69]
Lip	64	M	6 months	Not performed	Not performed	2014	[70]
Lip	71	M	11 months	Not performed	panCK (+), CD10 (+), vimentin (+), CK7 (−)	2014	[71]
Lip	60	M	5 months	Not performed	Not performed	2015	[72]
Tongue	70	M	Unknown	Not performed	EMA (+), vimentin (+), RCC (+), CD10 (+), HMWCK (+), CK7 (+)	2015	[73]
Gingiva	51	M	3 years	Not performed	CD10 (+), PAX8 (+)	2016	[74]
Gingiva	60	M	Same time	Not performed	Not performed	2016	[75]
Gingiva	31	F	4 years	Not performed	CD10 (+), vimentin (+), AE1/AE3 (+)	2016	[76]
Buccal mucosa	36	F	Same time	Not performed	CD10 (+), PAX8(+), vimentin (+), S100 (−)	2016	[77]
Gingiva	63	M	Same time	Not performed	vimentin (+), CD10 (+), CK (+), PAX8 (+), CK5 (−), CK6 (−), S100 (−)	2017	[78]
Gingiva	58	M	Unknown	Not performed	vimentin (+), CA (+), CK7 (−)	2017	[79]
Tongue	55	M	Same time	Not performed	PAX8 (+), CD10 (+), AE1/AE3 (+), CK7 (−), CK20 (−), TTF1 (−)	2017	[80]
Maxilla bone	54	M	Same time	Not performed	CD10 (+)	2018	[81]
Gingiva	51	M	Same time	Not performed	EMA (+), Ki-67 (+), HMB45 (−)	2018	[81]
Gingiva	78	F	Same time	Not performed	panCK (+), CK8/18 (+), PAX8 (+), CD10 (+), CA9 (+), CK19 (+), vimentin (+), EMA (+), CK20 (−), CK7 (−), p63 (−), p40 (−), CK5 (−), synaptophysin (−), c-kit (−), GATA3 (−), TTF1 (−), S100 (−), CDX-2 (−), calponin (−), calcitonin (−), EBER (−), HMB45 (−), PR (−), ER (−), CD31 (−)	2018	[82]
Tongue	51	M	Same time	Not performed	Not performed	2018	[18]
Gingiva	54	M	2 months	Not performed	CD10 (+), vimentin (+), HMB45 (−), S100 (−)	2018	[83]
Buccal mucosa	75	M	26 years	Not performed	CD10 (+), SMA (−), p63 (−), S100 (−)	2018	[21]
Buccal mucosa	59	M	Same time	Not performed	PAX8 (+)	2019	[84]
Tongue	54	M	Unknown	Not performed	Not performed	2019	[85]
Parotid gland	68	M	Same time	Not performed	Not performed	2019	[86]
Tongue	61	M	6 months	Not performed	vimentin (+), AE1/AE3 (+), CAM5.2 (+), CD10 (+), EMA (+), PAX8 (+), CK7 (−), CK20 (−), CD117 (−), p40 (−)	2020	[87]
Buccal mucosa	71	M	Same time	Not performed	vimentin (+), AE1/AE3 (+), CAM5.2 (+), CD10 (+), CK19 (+), CK7 (−), CK20 (−), EMA (−), CD117 (−), p40 (−)	2020	[87]
Mandible bone	56	F	Same time	Not performed	CD10 (+), PAX8 (+)	2020	[88]
Buccal mucosa	59	F	Same time	Not performed	AE1/AE3 (+), S100 (+), RCC (+), Melan-A (−), HMB45 (−), myogenin (−), chromogranin (−), synaptophysin (−)	2020	[89]
Gingiva	89	M	7 years	PAS (+)	CD10 (+), AE1/AE3 (+)	2020	[90]
Gingiva	53	M	Same time	Not performed	RCC (+)	2020	[91]

CEA: carcinoembryonic antigen, MSA: muscle-specific actin, HMB45: melanoma-associated antigen, LMWCK: low-molecular weight cytokeratins, PNRA: proximal nephron renal antigen, GCDFP: gross cystic disease fluid protein, TTF1: thyroid transcription factor-1, CA: carbonic anhydrase, PAX: Paired box protein, EBER: Epstein–Barr virus-encoded small RNAs, PR: progesterone receptor, ER: estrogen receptor.

## Data Availability

No new data were created or analyzed in this study. Data sharing is not applicable to this article.

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
