# Peer review of "Differential Diagnosis between Oral Metastasis of Renal Cell Carcinoma and Salivary Gland Cancer"

_diagnostics, 2021, doi:10.3390/diagnostics11030506_

Round 1
Reviewer 1 Report
This is a good paper for a frequent issue.
Can you suggest a minimum panel for a pathologist who is managing a lesion like this...?
Would reccomend to specify your search strategy in order to verify the papers presented and included.
Can you highlight the role of cytology in the cases included?
Woul recommend to cite these 2 following papers:
Appl Immunohistochem Mol Morphol. 2018 May/Jun;26(5):316-323. doi: 10.1097/PAI.0000000000000435.
Mol Diagn Ther. 2019 Oct;23(5):569-577. doi: 10.1007/s40291-019-00414-0.PMID: 31332726
Author Response
Reviewer #1
The authors are grateful to Reviewer #1 for the careful reading of our manuscript and the constructive and encouraging comments. We have revised our manuscript in response to Reviewer #1’s comments. Stated below are our replies to each of the comments made by Reviewer #1. We hope that these replies will meet the requirements of Reviewer #1 and the revised manuscript is acceptable for publication in the Journal.
Yours sincerely,
Yoshihiro Morita, D.D.S., Ph.D.
Comments from the Reviewer #1:
- Can you suggest a minimum panel for a pathologist who is managing a lesion like this...?
> Does "minimum pannel" mean a histopathological photograph or something like that? We have some histopathological photographs, but they have already been published in other journals and are copyrighted in their respective journals, so it is difficult to publish in this manuscript. Sorry….
- Would reccomend to specify your search strategy in order to verify the papers presented and included.
> We have added a new section, “5.1. Search strategy” and the statement, “A literature search was performed to retrieve previous studies describing the occurrence of oral metastases of renal cancer in the last 10 years. The following search items, combined with the Boolean term “AND,” were used to perform an electronic search in the PubMed database: oral metastasis, renal cell carcinoma, case report. Only English ones were selected from the literatures obtained by this search.” in the revised manuscript.
- Can you highlight the role of cytology in the cases included?
> We have added the statement, “Schmidt et al. mentioned that the clinical usefulness of fine-needle aspiration cytology for the diagnosis of salivary gland lesions is controversial in their systematic review paper.” in the “Differential Diagnosis of Clear Cell Tumors in the Oral Cavity” section in the revised manuscript.
- Woul recommend to cite these 2 following papers:
Appl Immunohistochem Mol Morphol. 2018 May/Jun;26(5):316-323. doi: 10.1097/PAI.0000000000000435.
> We have added the statement, “Moreover, CD13 and GATA-3 immunostains may serve as a diagnostic aid in differentiating subtypes of RCC” in the “4.3. Immunohistochemical Stains for Diagnosis” section in the revised manuscript.
Mol Diagn Ther. 2019 Oct;23(5):569-577. doi: 10.1007/s40291-019-00414-0.PMID: 31332726
> We have added the statement, “Also, in the genomic features, 9p loss appears a reliable and promising marker of patients with renal cell carcinoma associated with a worse prognosis.” in the “Renal Cell Carcinoma” section in the revised manuscript.
Reviewer 2 Report
line 25: In the section on introduction the authors state that in RCC one third of the case develop metastases, ... instead one third of case present metastases at initial diagnosis. Please check the data.
line 40: In the section on the epidemiology of renal cell cancer, the authors state that the peack of incidence is in patients aged between 60 and 70 years; in the introduction the age range is between of 30 and 60 years.
line 65: what's the meaning the sentence: "as a result, are characterized.." The sentence can be misunderstood. The hemorrhagic areas and sinusoidal component are histological features that allow to distinguish RCC from clear cell salivary tumors.
line 77: what's the meaning the sentence:"are almost always malignant in nature, they sometimes..." The sentence can be misunderstood: not sometimes, but they also include benign lesions.
line 170: Other immunostains may be usefull to diagnose RCC; please mention Pax8 and CAIX and discuss their usefulness in the differential diagnosis.
In the table 1 the PAS reaction is reported positive in acinic cell carcinoma, but it may be negative and if positive is diastase-resistent, unlike RCC. The authors mention the PAS diastase stain in the list of special stains and then no longer mention it.
Specify by which inclusion and exclusion criteria the articles are selected for the review
The references are too many, please reduce. Furthermore, some references are incorrectly written (e.g. references number 14 and 32)
Author Response
Reviewer #2
The authors are grateful to Reviewer #2 for the careful reading of our manuscript and the constructive and encouraging comments. We have revised our manuscript in response to Reviewer #2’s comments. Stated below are our replies to each of the comments made by Reviewer #2. We hope that these replies will meet the requirements of Reviewer #2 and the revised manuscript is acceptable for publication in the Journal.
Yours sincerely,
Yoshihiro Morita, D.D.S., Ph.D.
Comments from the Reviewer #2:
- line 25: In the section on introduction the authors state that in RCC one third of the case develop metastases, ... insteadone third of case present metastases at initial diagnosis. Please check the data.
> This means that metastases occur in one-third of all RCCs and half of those metastases are found after the initial diagnosis.
- line 40: In the section on the epidemiology of renal cell cancer, the authors state that the peack of incidence is in patients aged between 60 and 70 years; in the introduction the age range is between of 30 and 60 years.
> We have deleted the sentence, “The group most commonly affected by RCC is men between the ages of 30 and 60 years.” in line 22 in the revised manuscript.
- line 65:what's the meaning the sentence: "as a result, are characterized.." The sentence can be misunderstood. The hemorrhagic areas and sinusoidal component are histological features that allow to distinguish RCC from clear cell salivary tumors.
> We have changed the sentence, “and as a result, are characterized by…” to “but are characterized by…” in line 68 in the revised manuscript.
- line 77: what's the meaning the sentence:"are almost always malignant in nature, they sometimes..."The sentence can be misunderstood: not sometimes, but they also include benign lesions.
> We have changed the sentence, “are almost always malignant in nature, they sometimes include two types of benign lesions:” to “are often malignant in nature, they include two types of benign lesions:” in line 80 in the revised manuscript.
- line 170: Other immunostains may be usefull to diagnose RCC; please mention Pax8 and CAIX and discuss their usefulness in the differential diagnosis.
> We have added the statement, “Also, paired box 8 (PAX-8) and Carbonic anhydrase 9 (CA9, CAIX) expression are useful diagnostic markers for RCC. PAX-8 expression is detected in the primary tumor and distant sites. Clear cell RCC has lower PAX-8 expression and is less frequently positive compared with normal tissue and other histological types. Therefore, the lack of expression does not exclude a tumor of renal origin. CA9 is not expressed in normal renal tissue but is expressed in most clear cell RCC through HIF-1α accumulation driven.” in line 215 in the revised manuscript.
- In the table 1 the PAS reaction is reported positive in acinic cell carcinoma, but it may be negative and if positive is diastase-resistent, unlike RCC.The authors mention the PAS diastase stain in the list of special stains and then no longer mention it.
> We have added an asterisk to the PAS (+) listed in the Acinic cell carcinoma column of Table 1 and listed the “PAS diastase stain” in the footnote in the revised manuscript.
- Specify by which inclusion and exclusion criteria the articles are selected for the review
> We have added a new section, “5.1. Search strategy” and the statement, “A literature search was performed to retrieve previous studies describing the occurrence of oral metastases of renal cancer in the last 10 years. The following search items, combined with the Boolean term “AND,” were used to perform an electronic search in the PubMed database: oral metastasis, renal cell carcinoma, case report. Only English ones were selected from the literatures obtained by this search.” in the revised manuscript.
- The references are too many, please reduce. Furthermore, some references are incorrectly written (e.g. references number 14 and 32)
> We have revised, integrated and reduced the number of references.
Round 2
Reviewer 1 Report
Thanks to the authors.
I have no other remarks.
Reviewer 2 Report
Thanks, the authors responded comprehensively to the requested changes.
Please, just a minor change: in table 1, the minus sign in the special stains column of acinic cell carcinoma should must be placed inside the parenthesis.
I have no other remarks.